# Liquorice Toxicity: A Comprehensive Narrative Review

**DOI:** 10.3390/nu15183866

**Published:** 2023-09-05

**Authors:** Giovanna Ceccuzzi, Alessandro Rapino, Benedetta Perna, Anna Costanzini, Andrea Farinelli, Ilaria Fiorica, Beatrice Marziani, Antonella Cianci, Federica Rossin, Alice Eleonora Cesaro, Michele Domenico Spampinato, Roberto De Giorgio, Matteo Guarino

**Affiliations:** 1Department of Translational Medicine, St. Anna University Hospital of Ferrara, University of Ferrara, 44124 Ferrara, Italy; giovanna.ceccuzzi@unife.it (G.C.); alessandro.rapino@unife.it (A.R.); benedetta.perna@unife.it (B.P.); anna.costanzini@unife.it (A.C.); andrea.farinelli@edu.unife.it (A.F.); ilaria.fiorica@unife.it (I.F.); beatrice.marziani@unife.it (B.M.); antonella.cianci@unife.it (A.C.); federica.rossin@unife.it (F.R.); aliceeleonora.cesaro@edu.unife.it (A.E.C.); spmmhl@unife.it (M.D.S.); grnmtt@unife.it (M.G.); 2Department of Emergency, St. Anna University Hospital of Ferrara, University of Ferrara, 44124 Ferrara, Italy

**Keywords:** glycyrrhetinic acid, glycyrrhizin, liquorice, mortality, pseudo-hyperaldosteronism, toxicity

## Abstract

Background: Renowned since ancient times for its medical properties, liquorice is nowadays mainly used for flavoring candies or soft drinks. Continuous intake of large amounts of liquorice is a widely known cause of pseudo-hyperaldosteronism leading to hypertension and hypokalemia. These manifestations are usually mild, although in some cases may generate life-threatening complications, i.e., arrhythmias, muscle paralysis, rhabdomyolysis, and coma. In addition, liquorice has an important estrogenic-like activity. Methods: We summarized the current knowledge about liquorice and reviewed 104 case reports in both the English and Italian languages from inception to June 2023 concerning complications due to an excess of liquorice intake. Results: In contrast to most published data, female sex and old age do not appear to be risk factors. However, hypertension and electrolyte imbalance (mainly hypokalemia) are prevalent features. The detection of glycyrrhetinic acid in blood is very uncommon, and the diagnosis is essentially based on an accurate history taking. Conclusions: Although there is not a significant mortality rate, liquorice toxicity often requires hospitalization and therefore represents a significant health concern. Major pharmaceutical drug regulatory authorities should solicit public awareness about the potentially dangerous effects caused by excessive use of liquorice.

## 1. Introduction

Liquorice is the common name of a plant included in the *Fabaceae* family, specifically *Glycyrrhiza glabra* (Leguminosae). The main bioactive component is glycyrrhizin (GL), which is extracted from the roots. Renowned since ancient times for its medical properties, liquorice is nowadays mainly used for flavoring candies and soft drinks [1]. Nonetheless, its therapeutic properties are also well known (e.g., metabolic syndrome, asthma, recurrent oral ulcers, mental stress, and chronic liver diseases) [2]. Furthermore, a recent study highlighted its effectiveness in decreasing liver transaminase and blood lipids [3]. Because of the large, often exaggerated, consumption of GL, several cases (n = 104) of liquorice toxicity (LT) have been reported (Table 1). The most common pathogenetic mechanism related to LT is pseudo-hyperaldosteronism (PsA), characterized by typical features: arterial hypertension (AH), hypokalemia, and metabolic alkalosis (MA) [4]. Arrhythmias, skeletal muscle paralysis, rhabdomyolysis, and impaired consciousness are infrequent, although fatal, manifestations of LT [5]. The toxic threshold of GL defined by the World Health Organization is 100 mg/day. However, there is great variability depending on age, sex, associated drug use, and comorbidities (e.g., essential AH, chronic renal and hepatic failure) [6].

The present review Is aimed at providing a thorough appraisal about the multifaceted features of LT spanning from its pathogenetic, clinical, and diagnostic aspects to therapeutic strategies in order to aid physicians (in particular emergency ones) in their daily practice.

## 2. Search Strategy

PubMed, Scopus, and Google Scholar were searched from inception to June 2023. The search terms were “liquorice” OR “licorice” OR “Glycyrrhiza” OR “Glycyrrhizin” OR “glycyrrhetinic acid” AND “toxicity” OR “intoxication”. In addition, the analysis was expanded through a manual search of the references of the included studies and previous reviews. Non-Italian or non-English articles and irretrievable manuscripts were excluded from the analysis.

## 3. Historical Background, Source and Use of Liquorice

The genus name of *Glycyrrhiza* comes from the union of the ancient Greek words *glykos* (sweet) and *rhiza* (root). There are about 30 different naturally occurring species of *Glycyrrhiza* [5]. Most of the commercial liquorice is extracted from *Glycyrrhiza glabra*, a member of Leguminosae, growing especially in Southern Europe (varietas *typica*), the Middle East (varietas *violacea*), and Russia (varietas *glandulifera*). Other popular species are *Glycyrrhiza uralensis*, *Glycyrrhiza pallidiflora* (Chinese liquorice), and *Glycyrrhiza lepidota* (American liquorice) [102].

The first medical use of liquorice is witnessed in Assyrian and Egyptian prescriptions for treating bruises or swelling [103] and in Chinese culture, where it remains one of the most used medical herbs [2]. According to traditional Chinese medicine, liquorice is primarily used to intensify the properties of the other drugs and to better flavor herbal preparations. Moreover, it has been known to relieve or improve fatigue, debilitation, asthma, and phlegm [2]. In Europe, liquorice was fist mentioned by the Greek botanist Teophrastus (IV-III century B.C.) as “the Scinthian root” [104]. Its therapeutic properties were better described in the Roman period: Plinius suggested this plant as a remedy against infertility [105]. During the Middle Ages, the School of Salerno (VIII-IX century) fused Greco-Roman medical studies with Arabic knowledge and proposed liquorice to treat lung abnormalities (such as asthma and cough), gastrointestinal (GI) disorders (burning sensation, mouth ulceration and liver disorders), cardiovascular diseases (heart palpitations and draining), and skin and mucosae lesions [105].

Nowadays, liquorice is commonly used in several areas. Recognized as safe by the Food and Drug Administration, it is largely used for flavoring tobacco, food, and pharmaceutical drugs [1]. A broad range of biological activities have been described in literature, including anti-obesity, anti-diabetic [1,3,106], anti-inflammatory, and antioxidant properties [107]. Moreover, liquorice seems to exert a role in ameliorating allergic/inflammatory diseases of the skin [106], evoking ulcer healing, exerting a laxative effect [108,109,110], and providing hepatoprotective activity [109,111]. Furthermore, liquorice has been shown to have anticarcinogenic, antimicrobial, and antiviral activities, although the exact underlying mechanisms are not fully understood yet [106,110].

According to the Food and Drug Administration, approximately 90% of the consumption is related to smoking, while the dietary intake of Glycyrrhetinic acid (GA) is unremarkable [1]. However, the possibility of exceeding the safety threshold and developing signs/symptoms of toxicity can be linked to a daily overconsumption of readily available products (e.g., flavored candies or beverages). In this line, de-glycyrrhizinated liquorice has been recently produced in order to avoid LT as a possible side effect [5].

## 4. Chemical Composition

Commonly used liquorice products contain rhizomes and root extracts, as these parts of the plant are qualitatively and quantitatively richer in bioactive molecules than leaves, which are usually discarded by manufacturers [112,113]. As for most of the plant extract-based products, the phytochemical composition of liquorice extracts varies due to several factors, such as the genetic background of the plant variety, the environmental growing conditions, and the methods of liquorice processing [111,112]. Different extraction techniques and analytical methods have been developed and compared in order to determine the chemical composition of liquorice extracts and to improve the yield of the constituents of interest [114]. It has been estimated that about 40 to 50% of the dry weight of *Glycyrrhiza glabra* root extract consists of a bioactive mixture of water-soluble compounds [114].

Above 400 distinct molecules have been extracted by *Glycyrrhiza* species, with triterpene saponins, flavonoids, and free phenols being the most abundant ones [114]. While liquiritin and isoliquiritin flavonoids confer the yellow color to liquorice roots, the saponins GL, GA, and their triterpenoid derivatives have been identified as the most bioactive compounds of liquorice in vivo [109,115]. GL (also referred to as GA) consists of an 18β-GA molecule linked to a disaccharide formed by two β-D-glucuronic acid molecules [109,116] (Figure 1). Glucuronidase enzyme expressed by the plant or animal/human gut microbiota catalyzes the hydrolysis of GL to two GA pentacyclic triterpenoids, the epimers 18α-GA and 18β-GA [117], differing from each other due to their C18-H-, trans-, and cis-configuration. Evidence suggests that GA epimers have different activities in humans and animals [117,118].

## 5. Pharmacokinetics

The general population is usually exposed to liquorice derivatives after ingestion (i.e., foods, sweets, and supplements) and, to a much lower extent, to intravenous injection of GL, as reported in clinical studies investigating GL efficacy for chronic hepatitis treatment [115,120]. GL has a limited oral bioavailability, resulting in negligible blood levels after a single dose up to 1600 mg/kg [109]. After ingestion, GL undergoes pre-systemic hydrolysis to GA via specialized β-glucuronidase enzymes expressed by the intestinal microbiota (see Figure 2). In particular, *Eubacterium* spp. (strain GHL), *Ruminococcus* spp. PO1-3, and *Clostridium innocum* ES2406 evoke GL hydrolysis, whereas common β-glucuronidases expressed by *Escherichia coli* do not catalyze the GA production [121,122]. While in the plasma of rats, a small amount (4%) of GL was detectable only after a high oral dosage (i.e., 200 mg/kg), GL resulted undetectable in humans even after 100 to 800 mg of oral administration [120]. However, the identification of a minimal amount of GL (1.1–2.5% of the dose) in human urine suggests a minimal absorption of this molecule in the GI tract [123]. In parallel, after oral GL intake (10–480 mg/kg), GA has been reported to reach the highest plasma levels (Cmax) after 12 to 16 h (Tmax) in rats. A similar dynamic was described in humans (Tmax = 8–12 h) [120]. Thus, GL is fully metabolized and absorbed in the GA form.

The essential role of gut bacteria on the bioavailability of liquorice-active derivates was highlighted by the evidence that GA was absent in the plasma of germ-free rats treated with oral liquorice [120]. Moreover, the matrix of GL delivery (i.e., aqueous or pure extracts of liquorice roots) is proven to affect GA bioavailability [120]. After its release, GA is rapidly absorbed and conveyed to the liver almost completely combined (>99.9%) with serum albumin, occupying both specific and nonspecific bindings sites of this carrier protein. In the liver, the dehydroepiandrosterone sulfotransferase (also identified as sulfotransferase 2A1) transforms GA in 18β-glycyrrhetyl-3-O-sulfate (GA3S) subsequently expelled in the bile [120]. The canalicular multi-specific organic anion transporter (cMOAT) promotes the biliary excretion of GA derivatives [124]. Biliary metabolites of GA are then released into the GI tract and partially reconverted by enterobacteria into GA molecules that subsequently re-enter the blood circulatory system. Indeed, multiple absorption peaks in the GA plasma concentration–time curve were reported by pharmacokinetic studies on liquorice, showing a second maximal absorption peak approximately 20 h post-administration. Carrier proteins with restricted saturation capacity mediate the enterohepatic cycling of GL derivatives [120,124]. In conclusion, GA shows a biphasic pattern of elimination, in which the distribution in the central compartment is followed by a slow, dose-dependent phase of elimination. The enterohepatic cycling is a major determinant to understand the delay in the plasma clearance of GA [124]. However, conflicting data about the absorption rate have been reported by both preclinical and clinical studies [120]. These discrepancies in the quantitative analysis of GL active compound absorption are attributable to the inter-individual differences in GA blood levels after equal amount of liquorice intake and related GL hydrolyzation (by gut microbiota variety), permanence in the gut lumen (dependent on different GI transit), carrier protein expression, and liver function [109,120,124].

## 6. Biochemical Mechanism of Toxicity

LT mechanisms (summarized in Figure 2) are mainly triggered by the imbalance of cortisol and aldosterone pathways, leading to PsA. In physiological conditions, aldosterone and cortisol act as competitive ligands and activators of the mineralocorticoid receptors (MR) in the cortical collecting duct cells of the kidney because of their similar molecular structures and receptor affinity. However, cortisol activity on MR is limited by its degradation via 1β-hydroxysteroid-dehydrogenase type 2 (11β-HSD2), resulting in the production of cortisone, which is almost inactive due to the negligible affinity to MR [4,109,125]. Conversely, the aldosterone released by the adrenal glands exerts its functions in the kidney by mediating sodium (Na) and water reuptake along with potassium (K) elimination, thus preserving the acid-base equilibrium [126]. In particular, aldosterone controls the transcriptional modulation of Na-K exchangers, epithelial Na-channels, renal outer medullary K channels, aquaporins, and bicarbonate-chloride antiporters via MR activation. 

Liquorice derivatives inhibit 11β-HSD2, thus blocking cortisol conversion and eliciting MR activation by this hormone. In the renal glomeruli, this event induces an outsized aldosterone-like effect leading to an increase of Na reabsorption and K excretion, underlying the distinctive features of PsA (i.e., hypokalemia, AH, oedema, and MA) [4,109,125]. In adults, the 11β-HSD2 is also expressed in the heart, brain, and vasculature [125]. Liquorice metabolites evoke AH even by increasing arterial tone, improving contractile reaction to pressor hormones (via activation of the endothelin system) and decreasing the synthesis of endothelial nitric oxide [125]. Moreover, in vivo experiments suggest that liquorice retains a central hypertensive effect independent from the 11β-HSD2 inhibition. Indeed, the injection of GA into the rat brain boosted blood pressure (BP), with no changes in renal functions [125].

In vitro studies highlighted that both GL and GA inhibit 11β-HSD2, although the GA potency of inhibition is roughly 200 times greater than GL ones. In patients with liquorice-induced PsA, GA3S is the metabolite detected at the highest blood concentration, suggesting its pivotal role in LT onset [4]. 

The development of PsA activates the negative feedback control of the aldosterone release leading to BP increase and K level decrease. However, preclinical evidence suggests that aldosterone hepatic degradation might be restricted by GL or GA via the inhibition of the 5β-reductase and 3β-hydroxysteroid dehydrogenase, thus preserving the systemic levels of this hormone [127].

### Other Liquorice-Induced Side Effects

Further to PsA, GA was observed to affect sexual activity and reproduction in male rats, whereas it evoked a mild inhibition of androgenic hormones in humans [128,129]. In female rats, GA evoked estrogenic activity, as reflected by uterine response and vaginal opening [129].

In addition to GA, liquorice seems to have an estrogenic effect due to flavonoids and isoflavonoids. Liquiritigenin, a flavonoid component, exhibits good binding affinity for the bovine uterine estrogen receptor [129], while glabridin and glabrene mimic estrogen activity, maintaining calcium balance [128]. Moreover, GA is proved to induce miscarriages and should be avoided in subjects taking oral contraceptives, hydrocortisone, and prednisone. Finally, GA can have a contributory effect in the treatment of polycystic ovary syndrome [109,128]. 

In terms of GI function, isoliquitigenin, a flavonoid component of liquorice, may decrease bowel mobility, independent of cholinergic inhibition or adrenergic and/or nitrergic exacerbation [129]. 

## 7. Clinical Risk Factors for LT

The following paragraphs will detail the risk factors for developing LT.

### 7.1. Daily Dosage

The Scientific Committee on Food declared a GL consumption of 100 mg/day to be safe, based on studies involving human volunteers [6]. However, even lower doses may provoke critical manifestations according to different individual susceptibility to GA, as is widely reported in the literature [1,4,26,63,128,130,131,132]. Apart from one paper by Sigurjònsdòttir et al. [130], most studies showed no clear correlation between GA levels and severity of manifestations, such as, for example, degree of AH.

### 7.2. Age

Old age (over 65 years) may affect several processes of GL metabolism and there is an age-dependent decrease in 11-BHSD2 levels [4,5,131]. In addition, a reduction of glucocorticoid receptor concentration may cause a decrease in negative feedback in the hippocampus, with a resulting elevation in serum cortisol concentrations. This phenomenon may contribute to the inhibition of 11β-HSD2 [4,5,120,131]. Furthermore, the elderly are often affected by several comorbidities (and in particular delayed GI transit), which might increase the risk of LT. The mean age of patients included in this review was 53 ± 19 years, a finding that contrasts with published data supporting old age as a predisposing factor toward liquorice-induced PsA. Pediatric LT is an extremely rare condition with only two cases so far reported [19,72]. 

### 7.3. Sex

Female sex is more susceptible to LT [4]. Two main factors may play a role in LT: (i) a higher frequency of constipation, which affects the process of hydrolyzation of GL, and (ii) the estrogenic and antiandrogenic effects of liquorice, which influence calcium homeostasis, testosterone levels, and vaginal/uterine response (in animal studies) [129]. No sex differences were found in the development of AH, although an intrinsic difference in the renin angiotensin aldosterone system between females and males was demonstrated [132]. In particular, Sigurjònsdòttir et al. highlighted that serum aldosterone concentrations were markedly decreased in men vs. women after the administration of the same dose of liquorice [132]. Among the revised cases in this review, there was a slight male prevalence (57 out of 104), while major adverse cardiac events (MACEs) were more frequent in females (10 out of 13). 

### 7.4. Metabolism

Wide inter-individual variations in GA blood levels occur even at the same dose of liquorice intake, and this may be explained by differences in terms of gut microbiota-related GL hydrolyzation [109,120], actual content of GL in the intestinal lumen [109,120], and liver function [115]. 

As for the primary metabolism, the hydrolyzation ratio is also determined by the GL intestinal transit time: the slower it is, the more GL is hydrolyzed, thus resulting in a higher GA concentration [114]. This may explain why constipated patients experience LT more frequently than subjects with a normal bowel habit. 

Although GA cannot be eliminated by urine excretion, it can be absorbed by renal proximal tubular epithelial cells, becoming a substrate of organic anion transporters 1 and 3. Hypoalbuminemia related to altered liver function can increase the unbound fraction of GA and its availability to tubular epithelial cells, where 11β-HSD2 is expressed [133]. In addition, a significant inter-individual difference of liver enzymes and canalicular transport of glucuronides, which affect secondary metabolism of GA and its time of excretion, can be demonstrated [3]. In support of this concept, patients with chronic hepatitis and liver cirrhosis demonstrated 0.7 and 0.23 times lower plasma clearance, respectively, compared to healthy subjects [133]. 

### 7.5. Comorbidities and Concomitant Use of Other Medications

Since LT is related to the development of PsA, patients with AH, kidney disease, and hydro-electrolyte imbalance (i.e., hypokalemia, hypernatremia, and MA) are more prone to develop this toxicity [4,131]. In particular, patients with pre-existing cardiovascular disease are at higher risk of MACEs [131]. GI disorders with diarrhea and/or vomiting as well as renal losses could predispose to electrolyte imbalance [5]. Moreover, patients with polycystic ovary syndrome have a greater susceptibility to AH [125]; however, the mineralocorticoid and the mild antiandrogenic activity may mitigate the spironolactone side-effects in these patients [129]. 

The concomitant use of other medications can reduce or amplify the clinical feature of LT. Indeed, antihypertensive drugs may mask AH, while thiazide and loop diuretics increase the risk of severe hypokalemia. Finally, steroids increase the inhibition of 11β-HSD2, thereby promoting AH and other LT-related features. Conversely, K-sparing drugs (e.g., aldosterone blockers, angiotensin-converting enzyme inhibitors, and angiotensin receptor blockers) may prevent liquorice-induced hypokalemia [4].

## 8. Clinical Manifestations

The most described clinical picture of LT is linked to the aldosterone-like activity of liquorice, including hydro-electrolyte disorder with related ECG alterations, MA, skeletal muscle disorders, and AH (Table 1). 

### 8.1. Cardiovascular Disorders

Mild to severe AH with secondary organ impairment (i.e., hypertensive encephalopathy [19,26,51,72,73,83], hypertensive retinopathy [31,81], acute kidney injury [47,49,66,79], MACEs [11,24,25,42,43,45,54,59,88,97,99,100], pulmonary edema [22,94]) were commonly described. Sigurjònsdottir et al. showed a linear dose-response relationship between liquorice intake and BP increase [130]. Nevertheless, a pre-existent chronic antihypertensive therapy might hide AH. Only one paper described a patient with hypotension associated with supraventricular tachycardia secondary to liquorice-induced hypokalemia [76]. Severe K depletion is associated with typical ECG features (i.e., QT interval prolongation, U waves, and ST segment slight elevation in V1–V3). Few cases described hypokalemia-induced fatal arrhythmias leading to cardiac arrest in the absence of other established causes [25,42,45,59,86,88,97,99,100,134]. Liquorice was lethal in only two cases [88,97]. Hasegawa et al. reported a patient developing heart failure resembling dilated cardiomyopathy [24]. Acute heart failure with pulmonary edema has been described [22,94]. 

### 8.2. Muscle and Neurological Manifestations

In almost half of patients reported in the table (50 out of 104), muscle symptoms (i.e., fasciculation, myoclonus, fatigue) were the first complaint with mild to severe rhabdomyolysis. 

Among papers describing patients with altered mental status [19,37,72,101], Francini-Pesenti et al. and Ceccuzzi et al. reported an unconscious young woman with LT and daily oral contraceptive, confirming that this pharmacological combination is contraindicated, as previously described. In the other cases the unconsciousness was related to major adverse events, i.e., torsade de point [59,97,134], cardiac arrest [42,86,88], Brugada-like pattern [45], and ventricular tachycardia [99,100]. Neurological manifestations were quite frequently (27 out of 104) reported, such as flaccid paresis [29,33,40,45,77,84,87,96,101] and hypertensive encephalopathy microhemorrhages [51,83]. 

### 8.3. Others

GI symptoms are rare in LT and include abdominal pain, nausea, vomiting, and diarrhea [9,37,62,75,87]. 

Beyond hypertensive retinopathy, Dobbins et al. described five cases of retinal and occipital vasospasm, manifesting with a transient visual loss [28]. As described by Deutch et al., GA enhances the contractile response of smooth muscle, giving rise to an ocular migraine-like clinical phenotype, without headache [125].

Finally, Omar et al. reported a possible role of GA in thromboembolic events, but, as far as we know, only one case has been described [5,27].

Though liquorice is not acknowledged to affect the blood cell count in humans, Celik et al. described a patient with liquorice-induced thrombocytopenia [52]. Indeed, GL has been shown to cause platelet suppression in animal models [52,109].

Main clinical manifestations have been summarized in Table 2.

## 9. Diagnosis

Hyperaldosteronism/PsA induced by LT should be distinguished according to the various etiologies [4,97]. Hyperaldosteronism is an endocrine disorder occurring when the adrenal glands produce an excess of aldosterone and can be etiologically classified into primary and secondary forms [135,136]. Primary hyperaldosteronism is commonly due to Conn syndrome (i.e., a primary tumor of the adrenal gland) or bilateral (less frequently monolateral) adrenal hyperplasia, ectopic aldosterone-secreting tumors, aldosterone-producing adrenocortical carcinomas, and familiar hyperaldosteronism type 1 [135,136]. Secondary hyperaldosteronism results from an excessive activation of the renin-angiotensin-aldosterone system, related to renin-producing tumors, renal artery stenosis, or overload conditions with relative hypovolemia (e.g., left ventricular heart failure, cirrhosis with ascites, and pregnancy) [137]. PsA is characterized by hypokalemia, MA, and reduced renin level without an increase in aldosterone level. Primary causes include either gene abnormalities (i.e., MR mutations, Liddle’s syndrome, apparent mineralocorticoid excess syndrome, and congenital adrenal hyperplasia) or acquired mechanisms (i.e., hypercortisolism and LT) [138]. The main causes of hyperaldosteronism/PsA are summarized in Table 3 [101].

Although the clinical history is essential for LT diagnosis, liquorice overuse is often underreported or even overlooked. Indeed, the use of GL as sweetener for foods and drugs is widespread; therefore its unintentional intake may occur regardless the conscious consumption of candies [62,64]. Moreover, physicians should be aware of over-the-counter herbal mixtures commonly consumed to treat digestive symptoms or quit smoking [41], lose weight [9], or treat infertility [47]. To date, most of the herbal compounds are unlicensed by pharmaceutical drug regulatory authorities and the actual amount of active ingredients are neither listed nor standardized.

In only 4 out of 104 cases, GA was assayed in the blood [33,54,62,101], whereas in the other cases the diagnosis was performed only on anamnestic and clinical features. No evidence-based indications about serum/urinary GA tests have been reported. Nevertheless, this assessment should be recommended in unconscious patients with suspected LT in whom an accurate history-taking cannot be achieved.

## 10. Treatment and Prognosis

In almost all the reported cases, clinical manifestations improved with quitting liquorice consumption and intravenous hydro-electrolyte support. In 84 out of 104 cases, hospital admission was necessary. The LT prognosis is generally good, with complete symptom resolution in 30 days [86]. Since no direct antidote is available, the therapy only consists in correcting the hydro-electrolyte imbalance, reducing AH, and avoiding severe complications. The persistence of symptoms, such as AH, can be explained by large volume of GA distribution, its long half-life (12 h), and prolonged enterohepatic circulation. In addition, LT mechanisms generate a hormonal disorder through the amplification of gene transcription, which may need time to be completely restored [4,124,125,132]. Only two papers described cases of LT leading to death for MACEs [88,97].

## 11. Conclusions

The present review summarized the current knowledge about pathophysiology, predisposing risk factors, and clinical manifestations of LT, creating awareness of its potential hazard. Current evidence does not conclude about female sex and old age as established risk factors for LT. Indeed, data published to date lack systematic clinical reviews, since our paper is the first examining so many cases. Large-scale studies are needed to investigate this contention.

We highlighted the importance of a thorough and detailed anamnesis, including nutritional habits, during the diagnostic process. Moreover, physicians should interview patients (and their relatives) about possible use of over-the-counter herbal remedies to quit smoking and lose weight. People should receive adequate information by healthcare professionals about the risk of LT. On the other hand, physicians should be aware of the wide use of liquorice as supplement and replacement sweetener in dietary foods. In this setting, patients affected by cardiovascular diseases, chronic renal and hepatic failure and constipation are more prone to develop toxicity.

Despite the considerable worldwide use of liquorice and its potential toxicity, LT diagnosis remains largely clinical relying on accurate history taking revealing a high daily consumption. Since liquorice is often found in a mixture of substances (in food, drinks and herbal medicines above all), could be relevant to better define the poisoning in order to avoid future events. Only 4 cases out of 104 reported GA blood levels. This assessment might be advisable in unconscious patients in home anamnestic record is not obtainable. To date, most of the herbal compounds are unlicensed by pharmaceutical drug regulatory authorities and the actual amount of active ingredients are neither listed nor standardized.

Although death is not a frequent event in LT (2 cases out of 104), this condition requires hospitalization to prevent or treat life-threatening complications. In conclusion, although uncommon, LT is a dangerous condition which can occur due to widespread liquorice use. The knowledge of LT and how to manage it is of critical importance for most physicians, especially those practicing in an emergency setting.

## Figures and Tables

**Figure 1 nutrients-15-03866-f001:**
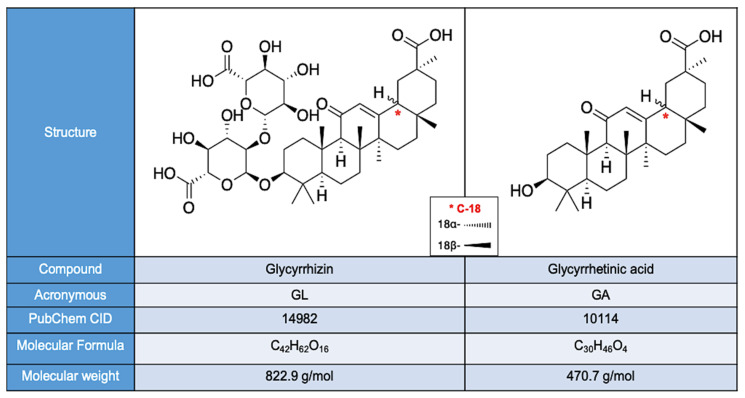
Chemical structure of GL and GA [119]. The wavy line expresses the bond which may vary in 3D-orientation among the two epimers.

**Figure 2 nutrients-15-03866-f002:**
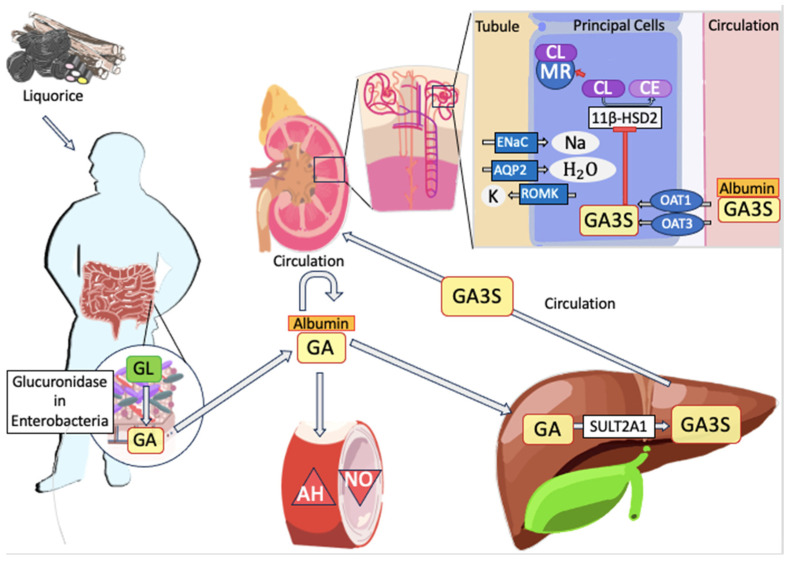
Schematic representation of metabolism pathway and toxicity mechanisms of liquorice and its derivatives. After oral intake, liquorice is converted to GL in the gut and this is hydrolyzed to GA by the β-glucuronidases of gut microbiota. GA is later absorbed in the blood stream and carried by serum albumin. In the liver, GA is converted to GA3S by SULT2A1. GA3S is excreted with the bile and enter the enterohepatic circulation. In tissues, GA and its metabolites inhibit 11β-HSD2 thus blocking the constitutive degradation of cortisol to cortisone. Available cortisol binds and activates MR cascade and their downstream effectors. In the kidneys, this event causes a potent increase in Na and H2O absorption combined with K excretion, via ROMKs, ENaCs and AQP2s channels. In blood vessels, GL metabolites boost contractile responses to pressor hormones and reduce endothelial NO production, concurring to AH. GL: Glycyrrhizin; GA: Glycyrrhetinic acid; GA3S: 18β-glycyrrhetyl-3-O-sulfate; SULT2A1: sulfotransferase 2A1; 11β-HSD2: 11-β-hydroxysteroid-dehydrogenase type II; CL: cortisol; CE: cortisone; MR: mineralocorticoid receptor; K: potassium; Na: sodium; H2O: water; ROMK: renal outer medullary potassium channel; ENaC: epithelial Na-channels; AQP2: acquaporine-2; OAT: organic anion transporter; NO: nitric oxide; AH: arterial hypertension.

**Table 1 nutrients-15-03866-t001:** Synopsis highlighting the main features of cases (n = 104) with LT published so far.

	Sex	Age	Electrolyte Abnormalities			ECG Alterations	Clinical Presentation					Aldosterone (pmol/L)	Cortisol (nmol/L)	Serum GA Levels (ng/mL)	Outcome	
			Hyper-natremia	Hypokalemia	Metabolic Alkalosis		Muscle Disorders	Neurological	MACEs	GI Disorders	Hypertension				Hospital Admission	Death
**Gross et al., 1966 [7]**	F	45	NO	YES	YES	YES	YES	NO	NO	NO	NO	N/A	N/A	N/A	YES	NO
**Conn et al., 1968 [8]**	M	58	YES	YES	YES	N/A	N/A	YES	NO	NO	YES	N/A	N/A	N/A	YES	NO
**Tourtelotte et al., 1970 [9]**	F	63	NO	YES	N/A	YES	YES	NO	NO	YES	YES	N/A	N/A	N/A	YES	NO
**Holmes et al., 1970 [10]**	M	63	YES	YES	N/A	YES	YES	YES	NO	NO	YES	N/A	N/A	N/A	YES	NO
**Bannister et al., 1977 [11]**	F	58	NO	YES	N/A	YES	YES	NO	YES	NO	YES	N/A	N/A	N/A	YES	NO
**Cumming et al., 1980 [12]**	F	70	NO	YES	NO	N/A	YES	NO	NO	NO	YES	N/A	N/A	N/A	YES	NO
**Nightingale et al., 1981 [13]**	F	25	NO	YES	YES	YES	YES	NO	NO	YES	NO	N/A	N/A	N/A	YES	NO
**Sundaram et al., 1981 [14]**	F	33	NO	YES	N/A	NO	NO	NO	NO	NO	YES	N/A	N/A	N/A	YES	NO
**Cibelli et al., 1984 [15]**	M	48	NO	YES	N/A	YES	YES	YES	NO	NO	YES	N/A	N/A	N/A	YES	NO
**Nielsen et al., 1984 [16]**	F	20	NO	YES	YES	YES	YES	NO	NO	NO	NO	N/A	N/A	N/A	YES	NO
**Farese et al., 1991 [17]**	M	70	NO	YES	N/A	N/A	YES	NO	NO	NO	NO	N/A	N/A	N/A	YES	NO
**Kageyama 1992 [18]**	M	54	NO	YES	N/A	N/A	NO	NO	NO	NO	YES	N/A	N/A	N/A	YES	NO
**Van Der Zwan 1993 [19]**	M	15	NO	NO	NO	YES	YES	YES	NO	NO	YES	N/A	N/A	N/A	YES	NO
**Heikens et al., 1995 [20]**	F	40	YES	YES	YES	N/A	NO	NO	NO	NO	YES	N/A	N/A	N/A	YES	NO
**Barella et al., 1997 [21]**	M	61	N/A	YES	N/A	YES	YES	NO	NO	NO	YES	95	N/A	N/A	YES	NO
**Chamberlain et al., 1997 [22]**	M	64	NO	YES	N/A	NO	NO	NO	NO	NO	N/A	N/A	N/A	N/A	YES	NO
**De Klerk et al., 1997 [23]**	F	21	NO	YES	N/A	N/A	NO	NO	NO	NO	YES	160	N/A	N/A	NO	NO
**De Klerk et al., 1997 [23]**	F	35	NO	YES	N/A	N/A	NO	NO	NO	NO	YES	80	N/A	N/A	NO	NO
**Hasegawa et al., 1998 [24]**	M	65	NO	YES	YES	YES	YES	NO	YES	NO	NO	<25	N/A	N/A	YES	NO
**Erikson et al., 1999 [25]**	F	44	NO	YES	N/A	YES	YES	NO	NO	NO	YES	4200	N/A	N/A	NO	NO
**Russo et al., 1999 [26]**	M	42	NO	YES	N/A	N/A	NO	YES	NO	NO	YES	180	N/A	N/A	YES	NO
**Russo et al., 1999 [26]**	M	46	NO	YES	YES	YES	NO	YES	NO	NO	YES	N/A	N/A	N/A	NO	NO
**Lozano et al., 2000 [27]**	F	34	NO	YES	N/A	YES	NO	NO	NO	NO	NO	68	14.5	N/A	YES	NO
**Dobbins et al., 2000 [28]**	M	62	N/A	N/A	N/A	N/A	N/A	N/A	N/A	N/A	N/A	N/A	N/A	N/A	NO	NO
**Dobbins et al., 2001 [28]**	M	76	N/A	N/A	N/A	N/A	N/A	N/A	N/A	N/A	N/A	N/A	N/A	N/A	NO	NO
**Dobbins et al., 2002 [28]**	F	26	N/A	N/A	N/A	N/A	N/A	N/A	N/A	N/A	N/A	N/A	N/A	N/A	NO	NO
**Dobbins et al., 2003 [28]**	M	65	N/A	N/A	N/A	N/A	N/A	N/A	N/A	N/A	N/A	N/A	N/A	N/A	NO	NO
**Dobbins et al., 2004 [28]**	M	39	N/A	N/A	N/A	N/A	N/A	N/A	N/A	N/A	N/A	N/A	N/A	N/A	NO	NO
**Elinav et al., 2003 [29]**	M	36	N/A	YES	N/A	YES	YES	YES	NO	NO	YES	N/A	N/A	N/A	YES	NO
**Lin S. et al., 2003 [30]**	M	76	YES	YES	YES	N/A	YES	YES	NO	NO	YES	N/A	N/A	N/A	YES	NO
**Hall et al., 2004 [31]**	F	62	N/A	N/A	N/A	N/A	NO	NO	NO	NO	YES	N/A	N/A	N/A	NO	NO
**Janse et al., 2005 [32]**	F	85	NO	YES	N/A	N/A	NO	YES	NO	NO	YES	30	N/A	N/A	YES	NO
**Van Den Bosch et al., 2005 [33]**	M	59	YES	YES	YES	N/A	YES	YES	NO	NO	YES	1.48	N/A	257	YES	NO
**Breidthardt et al., 2006 [34]**	N/A	67	NO	YES	N/A	N/A	NO	NO	NO	NO	YES	N/A	N/A	N/A	YES	NO
**Hamidon et al., 2006 [35]**	M	31	YES	NO	YES	N/A	YES	YES	NO	NO	YES	N/A	420	N/A	YES	NO
**Yasue et al., 2007 [36]**	F	93	NO	YES	YES	YES	YES	NO	NO	NO	YES	N/A	N/A	N/A	YES	NO
**Francini-Pesenti et al., 2008 [37]**	F	39	NO	YES	N/A	N/A	YES	YES	NO	YES	NO	15.08	N/A	N/A	YES	NO
**Mumoli et al., 2008 [38]**	M	55	N/A	YES	YES	YES	YES	NO	NO	NO	NO	60	N/A	N/A	YES	NO
**Sontia et al., 2008 [39]**	F	55	YES	YES	N/A	N/A	NO	NO	NO	NO	YES	31	N/A	N/A	NO	NO
**Tancevsky et al., 2008 [40]**	F	54	N/A	YES	N/A	YES	NO	YES	NO	NO	N/A	N/A	N/A	N/A	YES	NO
**Meltem et al., 2009 [41]**	M	21	NO	YES	NO	NO	YES	NO	NO	NO	NO	N/A	N/A	N/A	YES	NO
**Crean et al., 2009 [42]**	F	71	NO	YES	N/A	YES	NO	NO	YES	NO	NO	N/A	N/A	N/A	YES	NO
**Johns 2009 [43]**	F	49	NO	NO	NO	N/A	YES	NO	YES	NO	NO	N/A	N/A	N/A	NO	NO
**Murphy et al., 2009 [44]**	F	64	NO	YES	N/A	NO	NO	NO	NO	NO	YES	60	N/A	N/A	NO	NO
**Yorgun et al., 2010 [45]**	F	50	NO	YES	NO	YES	YES	YES	YES	NO	YES	N/A	N/A	N/A	YES	NO
**Katchanov et al., 2010 [46]**	M	70	NO	YES	YES	N/A	YES	NO	NO	NO	YES	75	N/A	N/A	YES	NO
**Velickovic-Radovanovic et al., 2010 [47]**	F	39	NO	YES	NO	YES	YES	NO	NO	YES	YES	14.28	N/A	N/A	YES	NO
**Støving et al., 2010 [48]**	F	18	NO	YES	NO	NO	NO	NO	NO	NO	NO	<25	N/A	N/A	YES	NO
**Kasap et al., 2010 [49]**	M	16	NO	YES	YES	YES	YES	NO	NO	YES	NO	1.50	N/A	N/A	YES	NO
**Imtiaz 2010 [50]**	M	49	NO	YES	N/A	NO	NO	NO	NO	NO	YES	<70	584	N/A	YES	NO
**van Beers et al., 2011 [51]**	M	49	NO	YES	YES	YES	NO	YES	NO	NO	YES	N/A	N/A	N/A	YES	NO
**Celik 2012 [52]**	M	45	N/A	YES	YES	N/A	NO	NO	NO	NO	NO	70	N/A	N/A	NO	NO
**van Noord et al., 2012 [53]**	M	62	N/A	YES	NO	NO	NO	NO	NO	NO	YES	N/A	N/A	N/A	YES	NO
**Kormann et al., 2012 [54]**	F	70	N/A	YES	N/A	YES	NO	NO	YES	NO	N/A	N/A	N/A	10000	YES	NO
**Ruiz-Granados et al. [55]**	F	51	NO	YES	YES	NO	NO	NO	NO	NO	YES	113	N/A	N/A	YES	NO
**Dehours et al., 2013 [56]**	M	35	N/A	N/A	N/A	YES	NO	NO	NO	NO	NO	N/A	N/A	N/A	NO	NO
**Flores-Robles et al., 2013 [57]**	F	47	NO	YES	NO	YES	NO	NO	NO	NO	YES	40	N/A	N/A	YES	NO
**Khan et al., 2013 [58]**	F	69	YES	YES	YES	N/A	NO	NO	NO	NO	YES	70	N/A	N/A	NO	NO
**Panduranga et al., 2013 [59]**	F	38	NO	YES	YES	YES	NO	NO	YES	YES	NO	N/A	N/A	N/A	YES	NO
**Kronborg-white 2013 [60]**	M	60	NO	YES	YES	N/A	YES	YES	NO	NO	YES	N/A	N/A	N/A	YES	NO
**Horwitz et al., 2014 [61]**	M	65	N/A	YES	YES	N/A	YES	NO	NO	NO	YES	N/A	N/A	N/A	YES	NO
**Cortini E. et al., 2014 [62]**	F	55	NO	YES	NO	YES	NO	YES	NO	YES	YES	N/A	N/A	63	YES	NO
**De Putter et al., 2014 [63]**	M	52	NO	YES	YES	N/A	YES	NO	NO	NO	YES	28	535	N/A	YES	NO
**Main et al., 2015 [64]**	N/A	21	YES	YES	N/A	N/A	YES	NO	NO	NO	NO	38	N/A	N/A	NO	NO
**koku et al., 2015 [65]**	M	57	NO	YES	N/A	YES	NO	NO	NO	NO	NO	30.88	N/A	N/A	YES	NO
**Danis et al., 2015 [66]**	M	49	NO	YES	YES	N/A	YES	NO	NO	NO	NO	N/A	N/A	N/A	YES	NO
**Machalke et al., 2015 [67]**	M	57	NO	YES	N/A	N/A	NO	NO	NO	NO	YES	11,000	4.25	N/A	YES	NO
**Dai et al., 2015 [68]**	M	66	n/a	YES	YES	N/A	NO	NO	NO	NO	YES	12.87	N/A	N/A	YES	NO
**Hataya et al., 2015 [69]**	M	81	NO	YES	YES	NO	YES	NO	NO	NO	YES	61.8	357	N/A	YES	NO
**Allcock et al., 2015 [70]**	F	45	NO	N/A	N/A	NO	NO	NO	NO	NO	YES	N/A	N/A	N/A	YES	NO
**Caravaca-Fontan et al., 2015 [71]**	M	15	N/A	YES	YES	YES	YES	NO	NO	YES	NO	43.8	236	N/A	YES	NO
**Tassinari et al., 2015 [72]**	M	10	NO	NO	N/A	N/A	NO	YES	NO	NO	YES	N/A	N/A	N/A	YES	NO
**O’Connell et al., 2016 [73]**	F	56	N/A	YES	N/A	N/A	NO	YES	NO	NO	YES	N/A	N/A	N/A	YES	NO
**Erkus et al., 2016 [74]**	M	57	N/A	YES	N/A	YES	NO	NO	NO	NO	NO	171.36	N/A	N/A	YES	NO
**Foster et al., 2017 [75]**	F	48	NO	YES	N/A	NO	NO	NO	NO	NO	YES	<100	N/A	N/A	YES	NO
**Foster et al., 2017 [75]**	M	51	YES	YES	N/A	N/A	YES	NO	NO	YES	YES	83	N/A	N/A	YES	NO
**Mehrtash et al., 2017 [76]**	M	22	NO	YES	N/A	YES	NO	NO	NO	NO	NO	N/A	N/A	N/A	NO	NO
**Sayiner et al., 2017 [77]**	M	43	NO	YES	YES	N/A	YES	YES	NO	NO	NO	N/A	N/A	N/A	YES	NO
**Varma et al., 2017 [78]**	F	70	NO	YES	N/A	NO	NO	NO	NO	NO	YES	N/A	N/A	N/A	NO	NO
**Zhong et al., 2017 [79]**	M	71	NO	YES	N/A	N/A	YES	NO	NO	NO	YES	N/A	N/A	N/A	YES	NO
**Gallacher et al., 2017 [80]**	M	65	N/A	YES	N/A	NO	YES	NO	NO	NO	YES	N/A	N/A	N/A	YES	NO
**Jing et al., 2018 [81]**	F	47	N/A	YES	N/A	N/A	YES	NO	NO	NO	YES	N/A	N/A	N/A	NO	NO
**Ramchandran et al., 2018 [82]**	M	45	YES	YES	YES	NO	YES	NO	NO	NO	YES	43.88	441.44	N/A	YES	NO
**Shin et al., 2019 [83]**	F	68	N/A	N/A	N/A	N/A	N/A	YES	N/A	N/A	YES	N/A	N/A	N/A	YES	NO
**Smedegaard et al., 2019 [84]**	F	43	NO	YES	N/A	YES	YES	YES	NO	NO	YES	N/A	N/A	N/A	YES	NO
**Falet et al., 2019 [85]**	M	84	N/A	YES	N/A	NO	NO	NO	NO	NO	YES	71	N/A	N/A	YES	NO
**Attou et al., 2020 [86]**	M	45	NO	YES	NO	YES	YES	NO	NO	NO	YES	106	N/A	N/A	YES	NO
**Attou et al., 2020 [86]**	F	30	NO	YES	YES	N/A	YES	YES	NO	NO	YES	N/A	N/A	N/A	YES	NO
**Benge et al., 2020 [87]**	F	74	NO	YES	N/A	NO	NO	YES	NO	YES	YES	12.87	N/A	N/A	YES	NO
**Edelman et al., 2020 [88]**	M	54	YES	NO	NO	YES	YES	YES	YES	NO	YES	51.48	N/A	N/A	YES	YES
**Kwon et al., 2020 [89]**	M	79	NO	YES	YES	YES	NO	NO	NO	NO	YES	24.58	N/A	N/A	YES	NO
**Awad et al., 2020 [90]**	M	62	NO	YES	NO	N/A	NO	NO	NO	NO	YES	12.87	N/A	N/A	YES	NO
**Petersen et al., 2020 [91]**	F	77	YES	YES	N/A	YES	NO	NO	NO	NO	YES	39	N/A	N/A	YES	NO
**Jing et al., 2020 [92]**	M	59	YES	YES	NO	N/A	NO	NO	NO	NO	YES	N/A	N/A	N/A	NO	NO
**Patel et al., 2021 [93]**	M	61	NO	YES	N/A	YES	YES	NO	NO	YES	YES	38.61	606.98	N/A	YES	NO
**McHugh J. et al., 2021 [94]**	F	79	NO	YES	YES	YES	YES	NO	YES	NO	YES	2600	N/A	N/A	YES	NO
**Khan et al., 2022 [58]**	M	56	NO	YES	YES	N/A	YES	NO	NO	NO	YES	N/A	N/A	N/A	YES	NO
**Shibata et al., 2022 [95]**	F	87	NO	YES	YES	YES	NO	NO	NO	NO	YES	92.5	N/A	N/A	YES	NO
**Bettini et al., 2022 [96]**	F	84	YES	YES	N/A	YES	YES	YES	NO	NO	YES	66.07	N/A	N/A	YES	NO
**Yoshida et al., 2022 [97]**	F	84	NO	YES	YES	YES	YES	NO	YES	NO	YES	N/A	N/A	N/A	YES	YES
**Blanpain 2023 [98]**	M	49	N/A	YES	N/A	NO	NO	NO	NO	NO	YES	N/A	N/A	N/A	YES	NO
**Han et al., 2023 [99]**	F	89	N/A	YES	N/A	YES	N/A	N/A	YES	NO	N/A	706	N/A	N/A	YES	NO
**Molina-Lopez et al., 2023 [100]**	M	95	YES	YES	N/A	YES	NO	NO	YES	YES	YES	12.87	N/A	N/A	YES	NO
**Ceccuzzi et al., 2023 [101]**	F	38	NO	YES	YES	YES	YES	YES	NO	YES	YES	24.45	817.26	115	YES	NO

Note. GI disorders: Abdominal tenderness, Nausea, Vomiting, Diarrhoea; Hypernatremia: Na Serum Concentration > 145 Mmol/L; Hypokalemia: K Serum Concentration < 3.6 Mmol/L; Metabolic Alkalosis: Serum pH > 7.45; Muscle Symptoms: Spams, Fasciculations, Myoclonus, Rhabdomyolysis, Fatigue; Neurological Symptoms: GCS < 15, Epileptiform Manifestations, Visual impairment due to retinic and occipital vasospasm, paresis, cerebral microhemorrhages, licorice induced myoclonus; MACEs: Major Cardiac Events; Hypertension: Systolic BP > 140 mmHg, Diastolic BP > 90 mmHg, cardiac arrest, heart failure and pulmonary edema, Cardiac arrhythmias and death due to QT prolongation, hypertensive encephalopathy, embolic ischemia, hypertensive retinopathy; Serum GA levels (ng/mL).

**Table 2 nutrients-15-03866-t002:** Main clinical manifestations of LT.

System	Clinical Manifestations
Cardiovascular disorder	Hypertension
	Cardiac arrest
	Heart failure and pulmonary edema
	Cardiac arrhythmias and death due to QT prolongation
	Hypertensive encephalopathy
	Embolic ischemia
	Hypertensive retinopathy
Neurological disorders	Liquorice induced myoclonus
	Cerebral microhemorrhages
	Paralysis
	Visual impairment due to retinic and occipital vasospasm
	Epileptiform manifestations
	Glasgow Coma Scale < 15
Electrolyte and renal abnormalities	Hypokalemia
	Hypernatremia
	Metabolic alkalosis
	Acute kidney injury
Muscular disorders	Spasm
	Rhabdomyolysis
	Elevated creatinine phospho-kinase
Gastrointestinal disorders	Abdominal tenderness
	Nausea
	Vomiting
	Diarrhoea

**Table 3 nutrients-15-03866-t003:** Main causes of hyperaldosteronism and pseudo-hyperaldosteronism [101].

HYPERALDOSTERONISM
**Low Renin (Angiotensin II Independent)**
*High Aldosterone* Primary hyperaldosteronism Aldosterone producing adenoma Unilateral/bilateral adrenal hyperplasia Adrenocortical carcinoma Glucocorticoid remediable aldosteronism
*Low Aldosterone* Pseudo-hyperaldosteronism - Acquired (hypercortisolism, liquorice toxicity, grapefruit) - Genetic (mutation of mineralocorticoids, Liddle’s syndrome, syndrome of apparent mineralocorticoid excess, congenital adrenal hyperplasia)
**High Renin (Angiotensin II dependent)**
*Secondary Hyperaldosteronism* Normotensive (Gittleman’s syndrome, Bartter’s syndrome) Hypertensive (renal artery stenosis, aortic coartaction, reninoma, congestive heart failure, liver cirrhosis with ascites)

## Data Availability

There are no data available for this paper.

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
