# Peer review of "Liquorice Toxicity: A Comprehensive Narrative Review"

_nutrients, 2023, doi:10.3390/nu15183866_

Round 1

Reviewer 1 Report

The topic is very important and has practical consequences. 

However, the paper should be improved before accepting it. 

1. The Introduction should contain a more comprehensive teview of licorice use, discussing the relevant medicinal uses (now the effect on transaminase is the first highlighted effect, although this is not the most important).

2. The whole topic should be introduced in a way that is more understandable. Eg. it is not a good practice to start the whole paper by mentioning GL, before discussing the constittuents-

3. Attention should be paid to explain the acronyms at their 1st mentioning (eg. LT) and the list of abbreviations should be completed (now many items are missing).

4. It is not clear what is the 100 mg/day WHO threshold: root? extract? DL? GA?

6. Plant and microbe names should be italicized.

7. varietas instead of variante (the latter is in Italian)

8. The first paragraph of section 3 should be rewritten by using botanical literature - the current reference is not primary literature.

9. It should be clear which compounds belong to the root - flavonoids are mentioned as flower components. The whole chemical part should be more comprehensive and clear.

10. Section 7.2      53 19 years (?)

11. What does the "Dosage of GA (ng/ml) mean in table 1? By the way, it would be essential to reportb the doses of licorice root/extract/GL/Ga ingested by the patients in reports. The dose/toxicity relatiuonship would be one of the most important part of the paper! This issues should be presented and discussed since this would have practical value!

The English of the manuscript should be improved. Although the overall quality is not inferior, some basic mistakes should be corrected (eg. molecules have been extracted by Glycyrrhiza), therefore I suggest to ask fpor the help of a native speaker.

Reviewer 2 Report

Liquorice is a famous medicinal plant and food additives, which is generally considered non-toxic. Of course, ingestion with high doses can also cause some symptoms. The manuscript reviewed 104 case reports in English and Italian language from inception to June 2023 concerning complications due to an excess of liquorice intake, providing some insights for the rational use of liquorice. Thus, I personally think the manuscript is meaningful and can be accepted for publication after minor revision.

1. Part 3, Paragraph 1: Plant Latin names should be italicized; the plant name of pallidiflora (Chinese liquorice) and lepidota (American liquorice) should be revised to Glycyrrhiza pallidiflora and Glycyrrhiza lepidota, respectively. The genus name of Glycyrrhiza cannot be omitted. Similar issues should be checked and avoided.

2. The structures of glycyrrhizin and glycyrrhetinic acid in Fig. 1 need to be redrawn using the Chemdraw software. In addition, the authors should label the C-18 position in the structure of glycyrrhetinic acid.

3.The manuscript extensively uses abbreviations, such as Glycyrrhizin (GL), Liquorice toxicity(LT), pseudo-hyperaldosteronism (PsA), arterial hypertension (AH), metabolic alkalosis (MA), gastrointestinal (GI), Glycyrrhetinic acid (GA), which increase the burden for readers. These specialized terms are not complex and thus are not recommended to be abbreviated in such way.

4. 3 of 24:“flowers, the saponins glycyrrhizin, GA and their derivatives triterpenoid have been identified as the most bioactive compounds of liquorice in vivo”. Note that GA isn’t a saponin.

5. Figure 2. is not easy to understand if we don’t read the legend and note.

6. For Section 7.1: The Scientific Committee on Food declared safe a consumption of 100 mg/day, based on studies involving human volunteers, what substance is limited to 100 mg/day? GA, GL or Liquorice?

Editing of English language is required.
